# HOMODISTIL: HOMOTOPIC TASK-AGNOSTIC DISTILLATION OF PRE-TRAINED TRANSFORMERS

**Chen Liang**[⋆,∗]**, Haoming Jiang**[⋄]**, Zheng Li**[⋄]**, Xianfeng Tang**[⋄]**, Bin Yin**[⋄] **& Tuo Zhao**[⋆]
[⋆]Georgia Institute of Technology, [⋄] Amazon
{cliang73,tourzhao}@gatech.edu,
{jhaoming,amzzhe,xianft,alexbyin}@amazon.com

## ABSTRACT

Knowledge distillation has been shown to be a powerful model compression approach to facilitate the deployment of pre-trained language models in practice. This paper focuses on task-agnostic distillation. It produces a compact pre-trained model that can be easily fine-tuned on various tasks with small computational costs and memory footprints. Despite the practical benefits, task-agnostic distillation is challenging. Since the teacher model has a significantly larger capacity and stronger representation power than the student model, it is very difficult for the student to produce predictions that match the teacher's over a massive amount of open-domain training data. Such a large prediction discrepancy often diminishes the benefits of knowledge distillation. To address this challenge, we propose Homotopic Distillation (HomoDistil), a novel task-agnostic distillation approach equipped with iterative pruning. Specifically, we initialize the student model from the teacher model, and iteratively prune the student's neurons until the target width is reached. Such an approach maintains a small discrepancy between the teacher's and student's predictions throughout the distillation process, which ensures the effectiveness of knowledge transfer. Extensive experiments demonstrate that HomoDistil achieves significant improvements on existing baselines.

## 1 INTRODUCTION

Pre-trained language models have demonstrated powerful generalizability in various downstream applications (Wang et al., 2018; Rajpurkar et al., 2016a). However, the number of parameters in such models has grown over hundreds of millions (Devlin et al., 2018; Raffel et al., 2019; Brown et al., 2020). This poses a significant challenge to deploying such models in applications with latency and storage requirements.

Knowledge distillation (Hinton et al., 2015) has been shown to be a powerful technique to compress a large model (i.e., teacher model) into a small one (i.e., student model) with acceptable performance degradation. It transfers knowledge from the teacher model to the student model through regularizing the consistency between their output predictions. In language models, many efforts have been devoted to task-specific knowledge distillation (Tang et al., 2019; Turc et al., 2019; Sun et al., 2019; Aguilar et al., 2020). In this case, a large pre-trained model is first fine-tuned on a downstream task, and then serves as the teacher to distill a student during fine-tuning. However, task-specific distillation is computational costly because switching to a new task always requires the training of a task-specific teacher. Therefore, recent research has started to pay more attention to *task-agnostic distillation* (Sanh et al., 2019; Sun et al., 2020; Jiao et al., 2019; Wang et al., 2020b; Khanuja et al., 2021; Chen et al., 2021), where a student is distilled from a teacher pre-trained on open-domain data and can be efficiently fine-tuned on various downstream tasks.

Despite the practical benefits, task-agnostic distillation is challenging. The teacher model has a significantly larger capacity and a much stronger representation power than the student model. As a result, it is very difficult for the student model to produce predictions that match the teacher's

---

∗Work done while interning at Amazon.

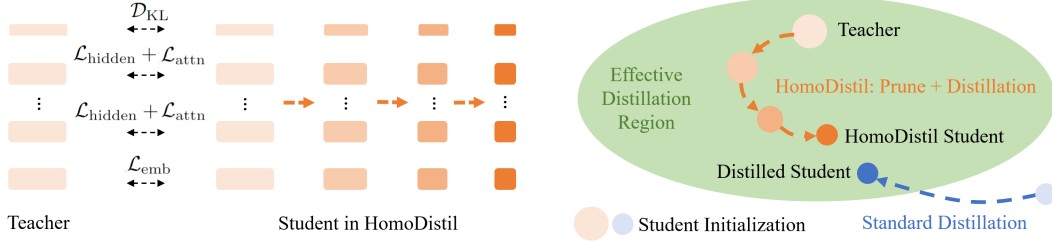

Figure 1: **Left**: In HomoDistil, the student is initialized from the teacher and is iteratively pruned through the distillation process. The widths of rectangles represent the widths of layers. The depth of color represents the sufficiency of training. **Right**: An illustrative comparison of the student's optimization trajectory in HomoDistil and standard distillation. We define the region where the prediction discrepancy is sufficiently small such that the distillation is effective as the *Effective Distillation Region*. In HomoDistil, as the student is initialized with the teacher and is able to maintain this small discrepancy, the trajectory consistently lies in the region. In standard distillation, as the student is initialized with a much smaller capacity than the teacher's, the distillation is ineffective at the early stage of training.

over a massive amount of open-domain training data, especially when the student model is not well-initialized. Such a large prediction discrepancy eventually diminishes the benefits of distillation (Jin et al., 2019; Cho & Hariharan, 2019; Mirzadeh et al., 2020; Guo et al., 2020; Li et al., 2021). To reduce this discrepancy, recent research has proposed to better initialize the student model from a subset of the teacher's layers (Sanh et al., 2019; Jiao et al., 2019; Wang et al., 2020b). However, selecting such a subset requires extensive tuning.

To address this challenge, we propose Homotopic Distillation (HomoDistil), a novel task-agnostic distillation approach equipped with iterative pruning. As illustrated in Figure 1, we initialize the student model from the teacher model. This ensures a small prediction discrepancy in the early stage of distillation. At each training iteration, we prune a set of least important neurons, which leads to the least increment in loss due to its removal, from the remaining neurons. This ensures the prediction discrepancy only increases by a small amount. Simultaneously, we distill the pruned student, such that the small discrepancy can be further reduced. We then repeat such a procedure in each iteration to maintain the small discrepancy through training, which encourages an effective knowledge transfer.

We conduct extensive experiments to demonstrate the effectiveness of HomoDistil in task-agnostic distillation on BERT models. In particular, HomoBERT distilled from a BERT-base teacher (109M) achieves the state-of-the-art fine-tuning performance on the GLUE benchmark (Wang et al., 2018) and SQuAD v1.1/2.0 (Rajpurkar et al., 2016a; 2018) at multiple parameter scales (e.g., 65M and $10 \sim 20$M). Extensive analysis corroborates that HomoDistil maintains a small prediction discrepancy through training and produces a better-generalized student model.

## 2 PRELIMINARY

### 2.1 TRANSFORMER-BASED LANGUAGE MODELS

Transformer architecture has been widely adopted to train large neural language models (Vaswani et al., 2017; Devlin et al., 2018; Radford et al., 2019; He et al., 2021). It contains multiple identically constructed layers. Each layer has a multi-head self-attention mechanism and a two-layer feed-forward neural network. We use $f(\cdot; \theta)$ to denote a Transformer-based model $f$ parameterized by $\theta$, where $f$ is a mapping from the input sample space $\mathcal{X}$ to the output prediction space. We define the loss function $\mathcal{L}(\theta) = \mathbb{E}_{x \sim \mathcal{X}}[\ell(f(x; \theta))]$, where $\ell$ is the task loss.[1]

### 2.2 TRANSFORMER DISTILLATION

**Knowledge Distillation** trains a small model (i.e., student model) to match the output predictions of a large and well-trained model (i.e., teacher model) by penalizing their output discrepancy. Specif-

---

[1]For notational simplicity, we will omit $x$ throughout the rest of the paper.

ically, we denote the teacher model as $f_t(\theta_t)$ and the student model as $f_s(\theta_s)$, and consider the following optimization problem:

$$\min_{\theta_s} \mathcal{L}(\theta_s) + \mathcal{D}_{\mathrm{KL}}(\theta_s, \theta_t), \tag{1}$$

where $\mathcal{D}_{\mathrm{KL}}(\theta_s, \theta_t)$ is the KL-Divergence between the probability distributions over their output predictions, i.e., $\mathrm{KL}(f_s(\theta_s)||f_t(\theta_t))$.

**Transformer Distillation.** In large Transformer-based models, distilling knowledge from only the output predictions neglects the rich semantic and syntactic knowledge in the intermediate layers. To leverage such knowledge, researchers have further matched the hidden representations, attention scores and attention value relations at all layers of the teacher and the student (Romero et al., 2014; Sun et al., 2019; 2020; Jiao et al., 2019; Hou et al., 2020; Wang et al., 2020b;a).

## 2.3 TRANSFORMER PRUNING

Pruning is a powerful compression approach which removes redundant parameters without significantly deteriorating the full model performance Han et al. (2015b;a); Paganini & Forde (2020); Zhu & Gupta (2017); Renda et al. (2020); Zafrir et al. (2021); Liang et al. (2021).

**Importance Score.** To identify the redundant parameters, researchers estimate the importance of each parameter based on some scoring metrics. A commonly used scoring metric is the sensitivity of parameters (Molchanov et al., 2017; 2019; Theis et al., 2018; Lee et al., 2019; Ding et al., 2019; Xiao et al., 2019). It essentially approximates the change in the loss magnitude when this parameter is completely zeroed-out (LeCun et al., 1990; Mozer & Smolensky, 1989). Specifically, we denote $\theta = [\theta_0, ..., \theta_J] \in \mathbb{R}^J$, where $\theta_j \in \mathbb{R}$ for $j = 1, ..., J$ denotes each parameter. We further define $\theta_{j,-j} = [0, ..., 0, \theta_j, 0, ..., 0] \in \mathbb{R}^J$. Then we define the sensitivity score as $S \in \mathbb{R}^J$, where $S_j \in \mathbb{R}$ computes the score of $\theta_j$ as

$$S_j = |\theta_{j,-j}^\top \nabla_\theta \mathcal{L}(\theta)|. \tag{2}$$

This definition is derived from the first-order Taylor expansion of $\mathcal{L}(\cdot)$ with respect to $\theta_j$ at $\theta$. Specifically, $S_j$ approximates the absolute change of the loss given the removal of $\theta_j$:

$$\theta_{j,-j}^\top \nabla_\theta \mathcal{L}(\theta) \approx \mathcal{L}(\theta) - \mathcal{L}(\theta - \theta_{j,-j}). \tag{3}$$

The parameters with high sensitivity are of high importance and should be kept (Lubana & Dick, 2020). Parameters with low sensitivity are considered redundant, and can be safely pruned with only marginal influence on the model loss. Other importance scoring metrics include the magnitude of parameters Han et al. (2015b) and the variants of sensitivity, e.g., movement score (Sanh et al., 2020), sensitivity score with uncertainty (Zhang et al., 2022), and second-order expansion of Eq 3 (LeCun et al., 1990).

**Iterative Pruning** gradually zeroes out the least important parameters throughout the training process. Specifically, given a gradient updated model $\theta^{(t)}$ at the $t$-th training iteration, iterative pruning methods first compute the importance score $S^{(t)}$ following Eq 2, then compute a binary mask $M^{(t)} \in \mathbb{R}^J$ as

$$M_j^{(t)} = \begin{cases} 1 & \text{if } S_j^{(t)} \text{ is in the top } r^{(t)} \text{ of } S^{(t)}, \\ 0 & \text{otherwise.} \end{cases} \quad \forall j = 1, ..., J. \tag{4}$$

where $r^{(t)} \in (0, 1)$ is the scheduled sparsity at the $t$-th iteration determined by a monotonically decreasing function of $t$. Then the model is pruned as $M^{(t)} \odot \theta^{(t)}$, where $\odot$ denotes the Hadamard product. Such a procedure is repeated through training.

**Structured Pruning.** Pruning the model in the unit of a single parameter leads to a highly sparse subnetwork. However, the storage and computation of sparse matrices are not often optimized on commonly used computational hardware. Structured pruning resolves this issue by pruning the model in the unit of a structure, e.g., a neuron, an attention head, or a feed-forward layer (Wang et al., 2019; Michel et al., 2019; Liang et al., 2021; Hou et al., 2020; Lagunas et al., 2021). To estimate the importance score of a structure, existing works compute the expected sensitivity with respect to the structure's output (Michel et al., 2019; Liang et al., 2021; Kim & Awadalla, 2020).

## 3 METHOD

We introduce Homotopic Distillation, as illustrated in Figure 1. Specifically, we initialize the student model from the teacher model. At each iteration, we prune the least important neurons from the student and distill the pruned student. We repeat such a procedure throughout the training process.

**Task-Agnostic Distillation.** We consider the following losses to optimize the student model: 1) The knowledge distillation loss as defined in Eq 1. In task-agnostic distillation, $\mathcal{L}$ is the loss for continual pre-training of the student model on the open-domain data, e.g., the masked language modeling loss for BERT, $\mathcal{L}_{\text{MLM}}$. 2) The Transformer distillation losses. Specifically, we penalize the discrepancy between the teacher's and the student's hidden representations at both the intermediate and embedding layers, and the attention scores at the intermediate layers. We denote the hidden representations at the $k$-th intermediate layer of the teacher and the student as $H_t^k \in \mathbb{R}^{|x| \times d_t}$ and $H_s^k \in \mathbb{R}^{|x| \times d_s}$, where $d_t$ and $d_s$ denote the hidden dimension and $|x|$ denotes the sequence length. The distillation loss of the hidden representations at the intermediate layers is defined as:

$$\mathcal{L}_{\text{hidn}}(\theta_s, \theta_t) = \sum_{k=1}^{K} \text{MSE}(H_t^k, H_s^k W_{\text{hidn}}^k).$$

Here $\text{MSE}(\cdot, \cdot)$ is the mean-squared error, and $W_{\text{hidn}}^k \in \mathbb{R}^{d_s \times d_t}$ is a randomly initialized and learnable linear projection that projects $H_s^k$ into the same space as $H_t^k$. Similarly, the distillation loss of the hidden representations at the embedding layer is defined as

$$\mathcal{L}_{\text{emb}}(\theta_s, \theta_t) = \text{MSE}(E_t, E_s W_{\text{emb}}),$$

where $E_t \in \mathbb{R}^{|x| \times d_t}$ and $E_s \in \mathbb{R}^{|x| \times d_s}$ are the hidden representations at the embedding layer and $W_{\text{emb}} \in \mathbb{R}^{d_s \times d_t}$ is for dimension matching. Finally, the attention distillation loss is defined as

$$\mathcal{L}_{\text{attn}}(\theta_s, \theta_t) = \sum_{k=1}^{K} \text{MSE}(A_t^k, A_s^k),$$

where $A_t^k \in \mathbb{R}^{|x| \times |x|}$ and $A_s^k \in \mathbb{R}^{|x| \times |x|}$ are the attention score matrices averaged by the number of heads at the $k$-th layer. These transformer distillation losses aim to capture the rich semantic and syntactic knowledge from the teacher's layers and improve the generalization performance of the student. In summary, the student is optimized based on the weighted sum of all losses, i.e.,

$$\mathcal{L}_{\text{total}} = \mathcal{L}_{\text{MLM}} + \alpha_1 \mathcal{D}_{\text{KL}} + \alpha_2 \mathcal{L}_{\text{hidden}} + \alpha_3 \mathcal{L}_{\text{emb}} + \alpha_4 \mathcal{L}_{\text{attn}}, \tag{5}$$

where $\alpha_1, \alpha_2, \alpha_3, \alpha_4 \geq 0$ are hyper-parameters.

**Iterative Neuron Pruning.** We initialize the student model from a pre-trained teacher model as

$$\theta_s^{(0)} = \theta_t.$$

At the $t$-th training iteration, we update the student model based on $\mathcal{L}_{\text{total}}$ defined in Eq 5 using an SGD-type algorithm, e.g.,

$$\theta_s^{(t)} \leftarrow \theta_s^{(t-1)} - \eta \nabla_{\theta_s^{(t-1)}} \mathcal{L}_{\text{total}}(\theta_s^{(t-1)}, \theta_t),$$

where $\eta$ is the step size. Then we compute the importance score for all parameters following Eq 2:

$$S_j^{(t)} = |\theta_{s\ j,-j}^{(t)\top} \nabla_{\theta_s^{(t)}} \mathcal{L}_{\text{total}}(\theta_s^{(t)}, \theta_t)| \quad \forall j = 1, ..., J. \tag{6}$$

For any weight matrix $W^{(t)} \in \mathbb{R}^{d_s^{\text{in}} \times d_s}$ in the student model, we denote its corresponding importance score as $S_W^{(t)} \in \mathbb{R}^{d_s^{\text{in}} \times d_s}$. We then define the importance score for individual columns as $N_W^{(t)} \in \mathbb{R}^{d_s}$, where

$$N_W^{(t)}{}_i = \|S_W^{(t)}{}_{[:,i]}\|_1 \quad \forall i = 1, ..., d_s. \tag{7}$$

Notice that the score is computed based on $\mathcal{L}_{\text{total}}$, which consists of both the distillation and training losses. This is to ensure that we only prune the columns whose removal would lead to the least increment in both the prediction discrepancy and the training loss.

We then compute the binary mask $M_W^{(t)} \in \mathbb{R}^{d_s^{\text{in}} \times d_s}$ associated with the weight matrix following Eq 4 as

$$M_{W\,[:,i]}^{(t)} = \begin{cases} \mathbf{1} & \text{if } N_{W\,i}^{(t)} \text{ is in the top } r^{(t)} \text{ of } N_W^{(t)}, \\ \mathbf{0} & \text{otherwise,} \end{cases} \quad \forall i = 1, ..., d_s, \tag{8}$$

where $r^{(t)}$ is the scheduled sparsity determined by a commonly used cubically decreasing function (Zhu & Gupta, 2017; Sanh et al., 2020; Zafrir et al., 2021):

$$r^{(t)} = \begin{cases} 1 & 0 \le t < t_i \\ r_f + (1 - r_f)\left(1 - \frac{t - t_i}{t_f - t_i}\right)^3 & t_i \le t < t_f \\ r_f & t_f \le t < T. \end{cases}$$

Here $r_f$ is the final sparsity, $T$ is the number of total training iterations and $0 \le t_i < t_f \le T$ are hyper-parameters. Such a schedule ensures that the sparsity is slowly increasing and the columns are gradually pruned. This prevents a sudden drop in the student's prediction performance, which effectively controls the expansion of prediction discrepancy.

Finally, we prune the weight matrix as $W^{(t)} \odot M_W^{(t)}$. We also prune the corresponding rows of the next weight matrix in the forward computation, including $\{W_{\text{hidn}}^k\}_{k=1}^K$ and $W_{\text{emb}}$. The same pruning procedure is applied to all weight matrices in the model. The complete algorithm is shown in Alg. 1.

---

**Algorithm 1** HomoDistil: Homotopic Distillation

1: **Input**: $\theta_t$: the teacher model. $T, t_i, t_f, r_f, \alpha_1, \alpha_2, \alpha_3, \alpha_4$: Hyper-parameters.
2: **Output**: $\theta_s^{(T)}$.
3: $\theta_s^{(0)} = \theta_t$.
4: **for** $t = 1, ..., T$ **do**
5:     Compute loss $\mathcal{L}_{\text{total}}$ following Eq 5.
6:     $\theta_s^{(t)} \leftarrow \theta_s^{(t-1)} - \eta \nabla_{\theta_s^{(t-1)}} \mathcal{L}_{\text{total}}$.
7:     Compute importance score $S^{(t)}$ following Eq 6.
8:     **for all** $W^{(t)} \in \theta_s^{(t)}$ **do**
9:         Compute importance score for individual columns, $N_W^{(t)}$, following Eq 7.
10:        Compute binary mask $M_W^{(t)}$ following Eq 8.
11:       $W^{(t)} \leftarrow W^{(t)} \odot M_W^{(t)}$.
12:     **end for**
13: **end for**

---

**Why do we impose sparsity requirements on individual weight matrices?** Traditional pruning imposes requirements on the global sparsity of the model instead of the local sparsity of the individual matrices. As a result, some matrices have much larger widths than the others. These wide matrices can be the memory bottlenecks for commonly used computational hardware. Furthermore, it requires re-configurations of the pre-defined model architectures in deep learning software packages to achieve the desired inference speedup. In contrast, controlling the local sparsity is more friendly to both hardware and software.

## 4 EXPERIMENTS

We evaluate HomoDistil on BERT-base (Devlin et al., 2018) on natural language understanding (NLU) and question answering tasks.

### 4.1 DATA

**Continual Pre-training.** We distill the student using the open-domain corpus for BERT pre-training (Devlin et al., 2018), i.e., Wikipedia [2], an English Wikipedia corpus containing 2500M words, and

---

[2]https://dumps.wikimedia.org/enwiki/

Toronto BookCorpus (Zhu et al., 2015), containing 800M words. We clean the corpus by removing tables, lists and references following BERT. We then pre-process the cleaned corpus by concatenating all sentences in a paragraph and truncating the concatenated passage by length of 128 following TinyBERT (Jiao et al., 2019). We tokenize the corpus with the vocabulary of BERT (30k).

**Fine-tuning.** We fine-tune the student model on both NLU and question answering tasks. For NLU tasks, we adopt the commonly used General Language Understanding Evaluation (GLUE) benchmark (Wang et al., 2018), which contains nine tasks, e.g., textual entailment, semantic similarity, etc. For question answering tasks, we adopt the SQuAD v1.1 and v2.0 (Rajpurkar et al., 2016a; 2018). Details about the datasets are deferred to Appendix A.1.

## 4.2 MODEL

We evaluate HomoDistil on pre-trained BERT-base (Devlin et al., 2018), which contains 12 Transformer layers with hidden dimension 768. BERT-base is pre-trained with masked language modeling and next sentence prediction tasks on Wikipedia and Toronto BookCorpus (16GB). We use BERT-base as the teacher model and as the initialization of the student model. We produce multiple student models at several sparsity ratios. Table 1 lists the architectures of the teacher and the student models.

Table 1: Architectures of the teacher and the student models.

| Model | Params (million) | | | $d^{\mathrm{hidn}}$ | $d^{\mathrm{ffn}}$ |
| | Embedding | Backbone | Total | | |
| --- | --- | --- | --- | --- | --- |
| BERT-base (Teacher) | 23.4 | 85.5 | 109 | 768 | 3072 |
| HomoBERT-base | 17.6 | 47.8 | 65 | 576 | 2304 |
| HomoBERT-small | 7.8 | 9.4 | 17.3 | 256 | 1024 |
| HomoBERT-xsmall | 7.3 | 8.3 | 15.6 | 240 | 960 |
| HomoBERT-tiny | 7.2 | 6.8 | 14.5 | 224 | 896 |

## 4.3 BASELINES

We compare HomoDistil with the state-of-the-art task-agnostic distillation baselines.[3] These methods initialize the student directly as the target size and fix its size during distillation. For example, to obtain a shallow model, the student is often initialized from a subset of teacher's layers.

**DistilBERT** (Sanh et al., 2019) considers the vanilla distillation by penalizing the final layer prediction discrepancy using Eq 1.
**TinyBERT-GD (General Distillation)** (Jiao et al., 2019) extends DistilBERT by exploiting the knowledge in the intermediate Transformer layers using Eq 5.
**MiniLM** (Wang et al., 2020b) penalizes the discrepancy between the queries-keys scaled dot product and values-values scaled dot product in the final layer self-attention module.
**MiniLMv2** (Wang et al., 2020a) extends MiniLM by encouraging the student to mimic the attention head relations of the teacher.

## 4.4 IMPLEMENTATIONS DETAILS

**Continual Pre-training.** For all experiments, we use a max sequence length of 128 and a batch size of 4k. We train the student model for $T = 28$k steps (3 epochs). We use Adam (Kingma & Ba, 2014) as the optimizer with $\beta = (0.9, 0.999)$, $\epsilon = 1 \times 10^{-6}$. We use a learning rate of $3 \times 10^{-4}$ for HomoBERT-base and $6 \times 10^{-4}$ for HomoBERT-small/xsmall/tiny. We adopt a linear decay learning rate schedule with a warmup ratio of 0.1. For distillation, we share all weights of $W_{\mathrm{hidn}}$ and $\{W_{\mathrm{emb}}^k\}_{k=1}^K$. We set $\alpha_1, \alpha_2, \alpha_3, \alpha_4$ to be 1 for all experiments. For importance score computation, we select neurons based on the exponential moving average of the importance score for stability. For pruning schedule, we set the initial iteration $t_i$ as 0 and select the final iteration $t_f$ from $\{0.5, 0.7, 0.9\} \times T$. Full implementation details are deferred to Appendix A.2.

**Fine-tuning.** We drop the masked language modeling prediction head and $W_{\mathrm{hidn}}$ and $\{W_{\mathrm{emb}}^k\}_{k=1}^K$ from the continual pre-training stage, and randomly initialize a task-specific classification head for

---

[3]We mainly compare with baselines that use BERT-base as the teacher model for a fair comparison. We also present a comprehensive comparison with task-specific distillation baselines in Appendix A.4.

the student model. For NLU tasks, we select the training epochs from $\{3, 6\}$, batch size from $\{16, 32\}$ and learning rate from $\{2, 3, 4, 5, 6, 7\} \times 10^{-5}$. For RTE, MRPC and STS-B, we initialize the student from a MNLI-fine-tuned student to further improve the performance for all baselines. For question answering tasks, we fine-tune the student for 2 epochs with a batch size of 12, and adopt a learning rate of $1 \times 10^{-4}$. For all tasks, we use Adam as the optimizer with with $\beta = (0.9, 0.999)$, $\epsilon = 1 \times 10^{-6}$. Full implementation details are deferred to Appendix A.3.

## 4.5 MAIN RESULTS

Table 2 show the fine-tuning results of HomoDistil on the GLUE development set. We report the median over five random seeds for all experiments in this paper [4]. HomoBERT-base consistently outperforms existing state-of-the-art baselines over six out of eight tasks, and achieves significant gains on MNLI, SST-2 and CoLA. The margins of gains become much more prominent for students with $10 \sim 20$M parameters: HomoBERT-tiny (14.1M) significantly outperforms TinyBERT$_{4 \times 312}$ (14.5M) by 3.3 points in terms of task-average score, and outperforms BERT-small, which is twice of the scale, by 1.0 point.

Table 3 show the fine-tuning results of HomoDistil on SQuAD v1.1/v2.0. All HomoBERT students outperform the best baseline, MiniLM$_3$ (17.3M), by over 3 points of margin on SQuAD v2.0. Especially, HomoBERT-xsmall (15.6M) obtains 3.8 points of gain.

Table 2: The accuracy of fine-tuning distilled BERT models on GLUE development set. The results of MiniLM$_{3/6}$ are reported from (Wang et al., 2020b). The rest are fine-tuned from the officially released checkpoints (Devlin et al., 2018; Sanh et al., 2019; Wang et al., 2020a; Jiao et al., 2019).

| Model | Params (million) | MNLI Acc | QQP Acc/F1 | QNLI Acc | SST-2 Acc | CoLA Acc | RTE Acc | MRPC Acc/F1 | STS-B P/S | Avg Score |
|---|---|---|---|---|---|---|---|---|---|---|
| BERT-base (Teacher) | 109 | 84.5/84.6 | 91.1/88.1 | 91.2 | 92.9 | 58.7 | 79.8 | 89.5/92.4 | 89.3/89.2 | 84.6 |
| DistilBERT$_6$ | 66 | 82.4/82.5 | 90.4/87.1 | 89.2 | 90.9 | 53.5 | 75.5 | 86.5/90.5 | 87.9/87.8 | 82.1 |
| TinyBERT$_6$-GD | 66 | 83.5/- | 90.6/- | 90.5 | 91.6 | 42.8 | 77.3 | 88.5/91.6 | 89.0/88.9 | 81.7 |
| MiniLM$_6$ | 66 | 84.0/- | 91.0/- | 91.0 | 92.0 | 49.2 | - | -/- | -/- | - |
| MiniLMv2$_6$ | 66 | 84.0/- | 91.1/- | 90.8 | 92.4 | 52.5 | 78.0 | 88.7/92.0 | 89.3/89.2 | 83.4 |
| HomoBERT-base | 65 | 84.2/84.3 | 91.2/87.9 | 90.7 | 92.7 | 55.9 | 77.6 | 89.0/91.9 | 89.5/89.2 | 83.8 |
| BERT-small | 28.6 | 78.8/78.9 | 89.9/86.5 | 87.0 | 88.2 | 36.1 | 70.8 | 85.8/90.1 | 87.7/87.7 | 78.0 |
| TinyBERT$_{3 \times 384}$-GD | 17.0 | 77.4/- | -/- | - | 88.4 | - | - | -/- | - | - |
| MiniLM$_3$ | 17.0 | 78.8/- | 88.8/85.0 | 84.7 | 89.3 | 15.8 | 66.4 | 81.9/88.2 | 85.4/85.5 | 73.9 |
| TinyBERT$_{4 \times 312}$-GD | 14.5 | 80.4/80.9 | 88.7/85.3 | 85.7 | 89.7 | 18.6 | 71.1 | 84.6/89.1 | 87.0/87.2 | 75.7 |
| HomoBERT-tiny | 14.1 | 81.2/81.3 | 89.9/86.6 | 87.8 | 90.1 | 37.0 | 70.8 | 87.3/90.7 | 87.6/87.5 | 79.0 |
| HomoBERT-xsmall | 15.6 | 81.5/81.8 | 90.0/86.7 | 88.0 | 90.3 | 40.8 | 71.5 | 87.7/91.0 | 88.3/88.0 | 79.7 |
| HomoBERT-small | 17.3 | 81.8/81.8 | 90.1/86.9 | 88.5 | 91.1 | 42.1 | 72.6 | 88.0/91.4 | 88.3/88.1 | 80.3 |

Table 3: The accuracy of fine-tuning distilled models on SQuAD v1.1/2.0 validation set. The results of TinyBERT$_3$-GD and MiniLM$_3$ are reported from (Wang et al., 2020b). The rest are fine-tuned from the officially released checkpoints (Devlin et al., 2018; Jiao et al., 2019).

| Model | Params (million) | SQuAD v1.1 EM/F1 | SQuAD v2.0 EM/F1 | Avg F1 |
|---|---|---|---|---|
| BERT-base (Teacher) | 109 | 81.7/88.9 | 73.4/76.7 | 82.8 |
| BERT-small | 28.6 | 72.5/81.5 | 61.3/64.8 | 73.2 |
| TinyBERT$_3$-GD | 17.0 | -/- | -/63.6 | - |
| MiniLM$_3$ | 17.0 | -/- | -/66.2 | - |
| TinyBERT$_4$-GD | 14.5 | 60.8/72.3 | 58.9/63.3 | 67.8 |
| HomoBERT-tiny | 14.1 | 75.5/84.1 | 66.1/69.5 | 76.8 |
| HomoBERT-xsmall | 15.6 | 76.2/84.5 | 66.5/70.0 | 77.2 |
| HomoBERT-small | 17.3 | 76.5/84.8 | 66.6/69.8 | 77.3 |

## 5 ANALYSIS

We verify that HomoDistil maintains a small prediction discrepancy throughout the distillation process, leading to a better-generalized student model.

---

[4]The standard deviations are reported in Appendix A.7.

## 5.1 HomoDistil Maintains a Small Prediction Discrepancy

Figure 2 shows the prediction discrepancy, $\mathcal{D}_{\mathrm{KL}}$, under different schedules of sparsity throughout the distillation process. When the student is directly initialized with a single-shot pruned subnetwork at the target sparsity (i.e., $t_f = 0$), the initial prediction discrepancy is large. In contrast, when the student is initialized with the full model and is iteratively pruned through longer iterations (i.e., $t_f = 0.5T, 0.7T$ and $0.9T$), the initial discrepancy is small. The discrepancy then gradually increases due to pruning, but the increment remains small due to distillation.

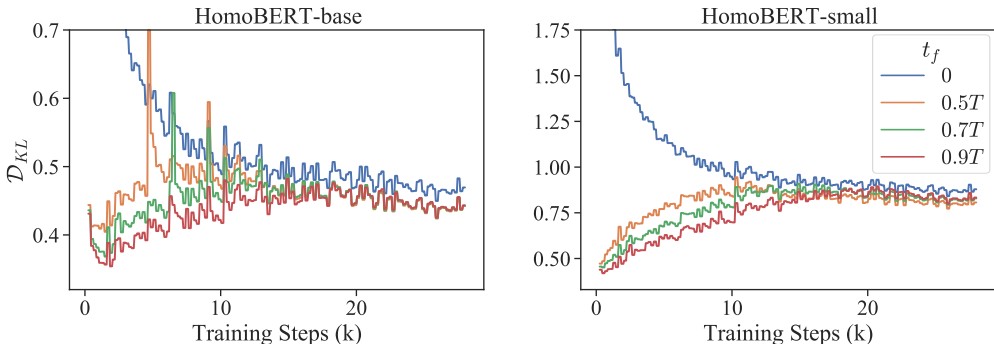

Figure 2: The prediction discrepancy during the distillation of HomoBERT models under different schedules of sparsity.

Figure 3 shows the accuracy of task-specific fine-tuning of the student distilled with different schedules of sparsity. The student that is initialized with the full model and is pruned iteratively achieves a significantly better generalization performance on the downstream tasks than the one initialized to be the target-size subnetwork.

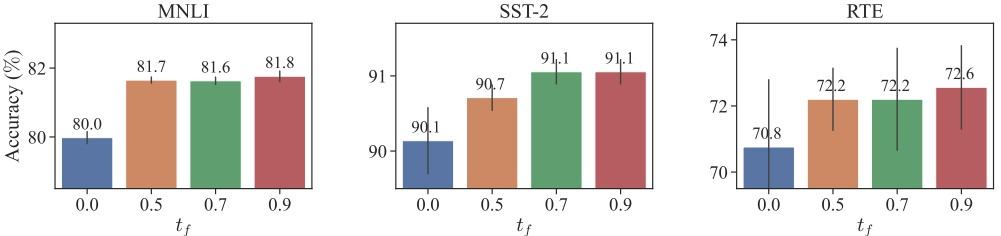

Figure 3: The accuracy of fine-tuning HomoBERT-small distilled with different schedules of sparsity on the development set of GLUE benchmark.

## 5.2 Distillation Benefits Iterative Pruning

Table 4 compares the student trained with and without distillation losses (i.e., $\mathcal{L}_{\mathrm{total}}$ defined in Eq 5 and $\mathcal{L}_{\mathrm{MLM}}$ only). The task-specific fine-tuning performance of the student trained with distillation losses consistently outperforms the one without distillation losses over multiple model scales. This suggests that teacher's knowledge is essential to recover the performance degradation due to pruning, and minimizing distillation loss is an important criteria to select important neurons.

## 5.3 Importance Metric Matters

Table 5 investigates the student performance under different importance metrics: 1) **Magnitude Pruning** (Han et al., 2015b), where $S_j = |\Theta_{j,-j}|$; 2) **Movement Pruning** (Sanh et al., 2020), where $S_j = \Theta_{j,-j}^{\top} \nabla_{\Theta} \mathcal{L}(\Theta)$; 3) **PLATON**(Zhang et al., 2022): $S_j = I_j \cdot U_j$, where $I_j$ is the sensitivity score as defined in Eq 2 and $U_j$ is the uncertainty estimation of $I_j$. For all methods, we use the exponential moving average of score for stability. Using sensitivity and PLATON as the importance score significantly outperforms the baseline. In contrast, the weight magnitude, which may not correctly quantify the neuron's contribution to the loss in the large and complex models, achieves

Table 4: The accuracy of fine-tuning HomoBERT models trained with and without distillation losses ("$\mathcal{L}_{\text{total}}$" and "$\mathcal{L}_{\text{MLM}}$") on the development set of GLUE benchmark.

| Importance Score | Params (million) | Loss Objective | MNLI Acc | SST-2 Acc | RTE Acc | Avg Score |
|---|---|---|---|---|---|---|
| BERT-base (Teacher) | 109 | - | 84.5/84.6 | 92.9 | 79.8 | 85.7 |
| HomoBERT-base | 65 | $\mathcal{L}_{\text{total}}$ | 84.2/84.3 | 92.7 | 77.6 | 84.8 |
| | | $\mathcal{L}_{\text{MLM}}$ | 82.8/83.0 | 91.6 | 75.1 | 83.2 |
| HomoBERT-small | 17.3 | $\mathcal{L}_{\text{total}}$ | 81.8/81.8 | 91.1 | 72.6 | 81.8 |
| | | $\mathcal{L}_{\text{MLM}}$ | 79.3/80.1 | 88.5 | 71.8 | 79.9 |
| HomoBERT-tiny | 14.5 | $\mathcal{L}_{\text{total}}$ | 81.2/81.3 | 90.1 | 70.8 | 80.7 |
| | | $\mathcal{L}_{\text{MLM}}$ | 78.7/79.5 | 87.7 | 69.7 | 78.7 |

only comparable performance to the baseline. Movement pruning, which is mainly designed for task-specific fine-tuning, diverges.

Table 5: The accuracy of fine-tuning HomoBERT-small pruned under different importance metrics on the development set of GLUE benchmark.

| Importance Score | Params (million) | MNLI Acc | SST-2 Acc | RTE Acc | Avg Score |
|---|---|---|---|---|---|
| BERT-base (Teacher) | 109 | 84.5/84.6 | 92.9 | 79.8 | 85.7 |
| TinyBERT$_{4\times32}$-GD | 17.0 | 80.4/80.9 | 89.7 | 71.1 | 80.4 |
| Magnitude(Han et al., 2015b) | 17.3 | 79.7/80.4 | 90.3 | 70.4 | 80.1 |
| Movement(Sanh et al., 2020) | 17.3 | Does not converge | | | |
| Sensitivity(LeCun et al., 1990) | 17.3 | 81.8/81.8 | 91.1 | 72.6 | 81.8 |
| PLATON(Zhang et al., 2022) | 17.3 | 81.6/81.9 | 90.6 | 73.6 | 81.9 |

## 6 DISCUSSION

**Combining pruning and distillation.** While we are the first work to combine pruning with distillation in task-agnostic setting, there have been similar explorations in task-specific setting. One stream of explorations first prune the model to the target size and then distill the subnetwork (Hou et al., 2020; Lagunas et al., 2021). In this case, pruning solely serves as an architecture selection strategy independent of distillation. Another stream simultaneously prunes and distills the model (Xu et al., 2021; Xia et al., 2022), which is more comparable to ours. The main differences are that they do not initialize the student with the teacher and often prune at a large granularity, e.g., a Transformer layer. In task-agnostic setting, however, an undesirable initialization and a large granularity will induce a huge discrepancy, which is difficult to minimize on large amount of open-domain data. Furthermore, after each layer pruning, the remaining layers need to match a different set of teacher layers to ensure the learning of comprehensive knowledge. However, suddenly switching the layer to learn from can be difficult on large amount of open-domain data. How to prune the student's height in task-agnostic setting remains an interesting open problem. A comprehensive comparison of these methods is deferred to Appendix A.5.

**Resolving prediction discrepancy.** Recent research has shown that distillation from a large teacher to a small student has only marginal benefits (Jin et al., 2019; Cho & Hariharan, 2019), mainly due to the large prediction discrepancy (Guo et al., 2020). Traditional solutions have resorted to introducing auxiliary teacher assistant models (Mirzadeh et al., 2020; Rezagholizadeh et al., 2021; Li et al., 2021), but training and storing auxiliary models can be memory and computational costly.

## 7 CONCLUSION

We propose a novel task-agnostic distillation approach equipped with iterative pruning – HomoDistil. We demonstrate that HomoDistil can maintain a small prediction discrepancy and can achieve promising benefits over existing task-agnostic distillation baselines.

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

# A  Appendix

## A.1  Data

**Continual Pre-training.** We use the same pre-training data as BERT: Wikipedia (English Wikipedia dump8 ; 12GB) and BookCorpus ((Zhu et al., 2015)) (6GB). We clean the corpus by removing tables, lists and references following BERT. We then pre-process the cleaned corpus by concatenating all sentences in a paragraph and truncating the concatenated passage by length of 128 following TinyBERT (Jiao et al., 2019)[5]. We tokenize the corpus with the vocabulary of BERT (30k).

**Fine-tuning.** GLUE is a commonly used natural language understanding benchmark containing nine tasks. The benchmark includes question answering (Rajpurkar et al., 2016b), linguistic acceptability (CoLA, Warstadt et al. 2019), sentiment analysis (SST, Socher et al. 2013), text similarity (STS-B, Cer et al. 2017), paraphrase detection (MRPC, Dolan & Brockett 2005), and natural language inference (RTE & MNLI, Dagan et al. 2006; Bar-Haim et al. 2006; Giampiccolo et al. 2007; Bentivogli et al. 2009; Williams et al. 2018) tasks. Details of the GLUE benchmark, including tasks, statistics, and evaluation metrics, are summarized in Table 16. SQuAD v1.1/v2.0 are the Stanford Question Answering Datasets (Rajpurkar et al., 2018; 2016a), two popular machine reading comprehension benchmarks from approximately 500 Wikipedia articles with questions and answers obtained by crowdsourcing. The SQuAD v2.0 dataset includes unanswerable questions about the same paragraphs.

## A.2  Continual Pre-training Implementations

Table 6 presents the hyper-parameter configurations for continual pre-training HomoBERT models on the open-domain data. We set the distillation temperature as 2. We empirically observe setting $t_f$ within the range of $0.5T \sim 0.9T$ can achieve similarly good downstream performances. Furthermore, we observe that different weight modules may prefer different $t_f$s: 1) it is better to finish the pruning of output projection matrices in the attention module, the feed-forward module and the embedding module early, because pruning them late will induce a large increment in distillation loss and the student performance is difficult to recover. 2) the student performance is less sensitive to the pruning of key and query projection matrices in the attention module and the input projection matrix in the feed-forward module, and can often easily recover. Based on this observation, we set $t_f = 0.5$ for the output projection matrices in the attention module, the feed-forward module and the embedding module. For key and query projection matrices in the attention module and the input projection matrix in the feed-forward module, we set $t_f = 0.9$. For other matrices, we set $t_f = 0.7$. This configuration brings around a small and consistent gain of $0.05 \sim 0.08$ on GLUE. The continual pre-training experiment runs for around 13 hours on 8 Nvidia A100 GPUs.

Table 6: Hyper-parameter configurations for task-agnostic distillation of HomoBERT models.

| Hyper-parameters | HomoBERT-base | HomoBERT-s | HomoBERT-xs | HomoBERT-tiny |
|---|---|---|---|---|
| Learning Rates | $3 \times 10^{-4}$ | $6 \times 10^{-4}$ | $6 \times 10^{-4}$ | $6 \times 10^{-4}$ |
| Batch Size | | 4000 | | |
| Training Epochs | | 3 | | |
| Learning Rate Decay | | Linear | | |
| Learning Rate Warmup | | 0.1 | | |
| Max Sequence Length | | 128 | | |
| Weight Decay | | 0.01 | | |
| Adam $\beta_1$ | | 0.9 | | |
| Adam $\beta_2$ | | 0.999 | | |
| Adam $\epsilon$ | | $1 \times 10^{-6}$ | | |
| Gradient Clipping | | None | | |
| $\alpha_1, \alpha_2, \alpha_3, \alpha_4$ | | 1 | | |
| $t_i$ | | 0 | | |

---

[5]https://github.com/yinmingjun/TinyBERT/blob/master/pregenerate_training_data.py

## A.3  FINE-TUNING IMPLEMENTATIONS

Table 7 presents the hyper-parameter configurations for fine-tuning HomoBERT models on the GLUE benchmark. We fine-tune the MRPC, RTE and STS-B from a fine-tuned MNLI student. All experiments are conducted on 1 Nvidia A100 GPU.

Table 7: Hyper-parameter configurations for fine-tuning HomoBERT models on the GLUE benchmark. "Epoch" refers to the total training epochs; we adopt early-stopping strategy in practice.

| Hyper-parameters | HomoBERT-base | HomoBERT-s | HomoBERT-xs | HomoBERT-tiny |
|---|---|---|---|---|
| Learning Rates | $\{2,3,4,5\} \times 10^{-5}$ | $\{4,5,6,7\} \times 10^{-5}$ | $\{6,7\} \times 10^{-5}$ | $\{6,7\} \times 10^{-5}$ |
| Batch Size | 16 for RTE and MRPC; 32 for the others. | | | |
| Training Epochs | 3 for MNLI and QNLI; 6 for the others. | | | |
| Learning Rate Decay | Linear | | | |
| Learning Rate Warmup | 0.05 | | | |
| Max Sequence Length | 128 | | | |
| Dropout of Task Layer | 0.1 | | | |
| Weight Decay | 0 | | | |
| Adam $\beta_1$ | 0.9 | | | |
| Adam $\beta_2$ | 0.999 | | | |
| Adam $\epsilon$ | $1 \times 10^{-6}$ | | | |
| Gradient Clipping | 1 | | | |

Table 8 presents the hyper-parameter configurations for fine-tuning HomoBERT models on the SQuAD v1.1/2.0. All experiments are conducted on 1 Nvidia A100 GPU.

Table 8: Hyper-parameter configurations for fine-tuning HomoBERT models on SQuAD v1.1/2.0.

| Hyper-parameters | HomoBERT-s/xs/tiny |
|---|---|
| Learning Rates | $1 \times 10^{-4}$ |
| Batch Size | 12 |
| Training Epochs | 2 |
| Learning Rate Decay | Linear |
| Learning Rate Warmup | 0.2 |
| Max Sequence Length | 384 |
| Dropout of Task Layer | 0 |
| Weight Decay | 0 |
| Adam $\beta_1$ | 0.9 |
| Adam $\beta_2$ | 0.999 |
| Adam $\epsilon$ | $1 \times 10^{-6}$ |
| Gradient Clipping | 1 |

## A.4  COMPARISON WITH TASK-SPECIFIC DISTILLATION METHODS

Table 9 compares HomoDistil with commonly used task-specific distillation baseline methods: PKD (Sun et al., 2019), BERT-of-Theseus (Xu et al., 2020), MixKD (Liang et al., 2020), DynaBERT (Hou et al., 2020), ProKT (Shi et al., 2021) and MetaDistil (Zhou et al., 2022). All baseline methods use a BERT-base fine-tuned on the target task as the teacher model, and a 6-layer pre-trained BERT-base as the initialization of the student model. The student model is then distilled with the target task data. As shown in the Table 9, HomoDistil demonstrates a prominent margin over the commonly used task-specific methods.

## A.5  A COMPARISON WITH "PRUNING+DISTILLATION" METHODS

We elaborate our discussion in Section 6 by comparing HomoDistil and the existing methods that combining pruning and distillation in more details. Table 10 and Table 11 present the detailed comparison among HomoDistil, DynaBERT (Hou et al., 2020), SparseBERT (Xu et al., 2021) and CoFi (Xia et al., 2022). We also list the major differences below:

**HomoDistil focuses on the task-agnostic setting.** In the task-specific setting, pruning incurs an inevitable loss of task-relevant pre-training knowledge that may not be present in the fine-tuning data. Therefore, existing works leave the word embeddings untouched (e.g., take up around 20 million parameters in BERT-base). In contrast, this problem does not exist in the task-agnostic

Table 9: The evaluation performance of HomoDistil and the commonly used task-specific distillation baseline methods on the GLUE development set. The results of PKD (Sun et al., 2019) and ProKT (Shi et al., 2021) are from our implementation. The rest results are from the original papers.

| Model | Params (million) | MNLI-m/mm Acc | QQP Acc | QNLI Acc | SST-2 Acc | CoLA Acc | RTE Acc | MRPC Acc | STS-B Spearman | Avg Score |
|---|---|---|---|---|---|---|---|---|---|---|
| $PKD_6$ (Sun et al., 2019) | 66 | 81.3/- | 88.4 | 88.4 | 91.3 | 45.5 | 66.5 | 85.7 | 86.2 | 79.2 |
| BERT-of-Theseus$_6$ (Xu et al., 2020) | 66 | 82.3/- | 89.6 | 89.5 | 91.5 | 51.1 | 68.2 | 89.0 | 88.7 | 81.2 |
| $MixKD_6$ (Liang et al., 2020) | 66 | 82.5/- | 90.8 | 88.8 | 92.1 | - | 67.9 | 84.1 | - | - |
| $DynaBERT_6$ (Hou et al., 2020) | 66 | 83.7/84.6 | 91.1 | 90.6 | 92.7 | 54.6 | 66.1 | 85.0 | 88.6 | 81.6 |
| $ProKT_6$ (Shi et al., 2021) | 66 | 82.8/83.2 | 90.9 | 89.7 | 91.3 | 54.3 | 68.4 | 86.3 | 88.9 | 81.6 |
| $MetaDistil_6$ (Zhou et al., 2022) | 66 | 83.5/83.8 | 91.0 | 90.4 | 92.3 | 58.6 | 69.4 | 86.8 | 89.1 | 82.6 |
| HomoBERT-base | 65 | 84.2/84.3 | 91.2 | 90.7 | 92.7 | 55.9 | 77.6 | 89.0 | 89.2 | 83.8 |

Table 10: Comparison of HomoDistil and the existing "Pruning+Distillation" methods from the distillation perspective.

| Method | Distillation Setting | Teacher Model | Student Initialization |
|---|---|---|---|
| DynaBERT (Hou et al., 2020) | Task-specific | Fine-tuned weights | Pruned, pre-trained weights |
| CoFi (Xia et al., 2022) | Task-specific | Fine-tuned weights | Pre-trained weights |
| SparseBERT (Xu et al., 2021) | Task-specific | Fine-tuned weights | Pre-trained weights |
| HomoDistil | Task-agnostic | Pre-trained weights | Pre-trained weights |

Table 11: A comparison of HomoDistil and the existing "Pruning+Distillation" methods from the pruning perspective.

| Method | Pruning Setting | Pruning Criterion | Pruning Granularity | Controllable Layer Width |
|---|---|---|---|---|
| DynaBERT | First prune then distill | Head and FFN sensitivity | Head, FFN | No |
| CoFi | Prune while distill | $\ell_0$ regularization | Layer, head, FFN, weight | No |
| SparseBERT | Prune while distill | Weight magnitude | Weight | No |
| HomoDistil | Prune while distill | Column sensitivity | Row, column | Yes |

setting. This allows us to prune the word embeddings and produce a smaller model more suitable for edge devices (e.g., around 15 million parameters). Furthermore, a task-specific model needs to be specifically pruned for each individual task, while a task-agnostic model can be fine-tuned for any task with a low cost.

**HomoDistil initializes the student with the teacher.** To maintain a small discrepancy in the early stage, HomoDistil initializes the student with the teacher. In contrast, DynaBERT initializes the student with a target-size subnetwork. SparseBERT and CoFi initialize the student with pre-trained weights while the teacher with fine-tuned weights.

**HomoDistil simultaneously prunes and distills and allows interactions between them.** To maintain a small discrepancy throughout distillation, HomoDistil prunes based on the sensitivity to make the pruning operation "distillation-aware". Specifically, HomoDistil selects the columns and rows to prune based on their contributions to the distillation loss. In contrast, DynaBERT treats pruning and distillation as two independent operations by first pruning then distilling the subnetwork. SparseBERT prunes based on the weight magnitude without considering the influence on the distillation loss.

**HomoDistil prunes rows and columns.** The granularity of rows and columns is sufficiently small to control the increment in discrepancy while maintaining the practical benefits of structured pruning.

**HomoDistil can control the layer width.** HomoDistil enforces a local sparsity constraint for each matrix, producing a model with consistent width in each layer. In contrast, SparseBERT and CoFi have no control over the layer width, which might result in wide matrices as the memory bottlenecks.

Table 12 shows the evaluation performance of HomoDistil, CoFi and SparseBERT on the GLUE benchmark (DynaBERT results are presented in Table 9). We can see that HomoDistil achieves a noticeable gain over CoFi and a comparable performance with SparseBERT with nearly half of their sizes.

Table 12: The performance comparison with the existing "Pruning+Distillation" methods. All results are reported by their papers.

| Model | Params (million) | MNLI-m/mm Acc | QQP Acc | QNLI Acc | SST-2 Acc | CoLA Acc | RTE Acc | MRPC Acc | STS-B Spearman | Avg Score |
|---|---|---|---|---|---|---|---|---|---|---|
| CoFi$_{5\%}$ (Xia et al., 2022) | 28.4 | 80.6/- | 90.1 | 86.1 | 90.6 | 35.6 | 64.7 | 82.6 | 83.1 | 76.7 |
| SparseBERT$_{5\%}$ (Xu et al., 2021) | 28.4 | -/- | - | 90.6 | - | 52.1 | 69.1 | 88.5 | - | - |
| HomoBERT-xsmall | 15.6 | 81.5/81.8 | 90.0 | 88.0 | 90.3 | 40.8 | 71.5 | 87.7 | 88.0 | 79.7 |

## A.6 COMPUTATIONAL COSTS

Table 13 compares the computational costs of HomoDistil and the baseline methods during inference. We profile the inference time and the number of FLOPs (embedding excluded) during the forward pass using the *profiler* package released by pytorch [6]. We conduct the measurements on the GLUE development set with a batch size of 128 and a maximum sequence length of 128 on one Nvidia A100 GPU. We compute the averaged time and FLOPs over all batches. The speedup is computed with respect to BERT-base. For a fair comparison, we only compare with compact models.

As can be observed, HomoDistil achieves some inference speedup and FLOPs reduction, but not as much as the other models under a similar parameter budget. This is because HomoDistil allocates a higher budget to the backbone parameters and a lower budget to the embedding parameters. However, we remark that HomoDistil achieves a better accuracy and enjoys the same (or more) storage benefits than the distilled (or structured pruned) models.

Table 13: The inference speedup and the number of FLOPs (embedding excluded) of HomoDistil and the baseline methods. The speedup is computed with respect to BERT-base.

| Model | Params (million) | Inference Speedup | # FLOPs (non-embedding) |
|---|---|---|---|
| BERT-base | 109 | 1.00× | 1.00× |
| DistilBERT$_6$ | 66 | 1.98× | 0.50× |
| TinyBERT$_6$-GD | 66 | 1.98× | 0.50× |
| MiniLMv1$_6$ | 66 | 1.98× | 0.50× |
| MiniLMv2$_6$ | 66 | 1.98× | 0.50× |
| HomoBERT-base | 65 | 1.30× | 0.56× |
| BERT-small | 28.6 | 4.77× | 0.15× |
| CoFi$_{5\%}$ | 28.4 | 5.16× | 0.05× |
| TinyBERT$_{3\times384}$-GD | 17.0 | 7.34× | 0.06× |
| MiniLMv1$_3$ | 17.0 | 7.34× | 0.06× |
| TinyBERT$_{4\times312}$-GD | 14.5 | 6.28× | 0.07× |
| HomoBERT-small | 17.1 | 2.40× | 0.11× |
| HomoBERT-xsmall | 15.6 | 2.51× | 0.10× |
| HomoBERT-tiny | 14.1 | 2.55× | 0.09× |

## A.7 STATISTICS OF EXPERIMENTAL RESULTS

All experimental results of HomoDistil presented in this paper are the median of five random seeds. Table 14 and Table 15 show the standard deviations of the experimental results on the GLUE benchmark (Table 2) and on the SQuAD v1.1/2.0 datasets (Table 3), respectively.

Table 14: The standard deviation of the experimental results on GLUE development set in Table 2.

| Model | MNLI-m/mm Acc | QQP Acc/F1 | QNLI Acc | SST-2 Acc | CoLA Acc | RTE Acc | MRPC Acc | STS-B P/S |
|---|---|---|---|---|---|---|---|---|
| HomoBERT-base | 0.13/0.20 | 0.09/0.12 | 0.34 | 0.24 | 1.72 | 0.93 | 1.17/0.83 | 0.15/0.18 |
| HomoBERT-small | 0.23/0.14 | 0.08/0.20 | 0.14 | 0.27 | 2.49 | 1.29 | 0.25/1.43 | 0.23/0.25 |
| HomoBERT-xsmall | 0.14/0.12 | 0.08/0.10 | 0.24 | 0.61 | 1.32 | 1.29 | 0.95/0.66 | 0.27/0.28 |
| HomoBERT-tiny | 0.16/0.29 | 0.11/0.13 | 0.29 | 0.16 | 1.26 | 1.26 | 1.05/0.62 | 0.19/0.22 |

---

[6]https://pytorch.org/tutorials/recipes/recipes/profiler_recipe.html

Table 15: The standard deviation of the experimental results on SQuAD v1.1/2.0 in Table 3.

| Model | SQuAD v1.1 EM | SQuAD v1.1 F1 | SQuAD v2.0 EM | SQuAD v2.0 F1 |
|---|---|---|---|---|
| HomoBERT-small | 0.24 | 0.19 | 0.37 | 0.37 |
| HomoBERT-xsmall | 0.15 | 0.19 | 0.62 | 0.59 |
| HomoBERT-tiny | 0.29 | 0.31 | 0.78 | 0.72 |

Table 16: Summary of the GLUE benchmark.

| Corpus | Task | #Train | #Dev | #Test | #Label | Metrics |
|---|---|---|---|---|---|---|
| Single-Sentence Classification (GLUE) | | | | | | |
| CoLA | Acceptability | 8.5k | 1k | 1k | 2 | Matthews corr |
| SST | Sentiment | 67k | 872 | 1.8k | 2 | Accuracy |
| Pairwise Text Classification (GLUE) | | | | | | |
| MNLI | NLI | 393k | 20k | 20k | 3 | Accuracy |
| RTE | NLI | 2.5k | 276 | 3k | 2 | Accuracy |
| QQP | Paraphrase | 364k | 40k | 391k | 2 | Accuracy/F1 |
| MRPC | Paraphrase | 3.7k | 408 | 1.7k | 2 | Accuracy/F1 |
| QNLI | QA/NLI | 108k | 5.7k | 5.7k | 2 | Accuracy |
| Text Similarity (GLUE) | | | | | | |
| STS-B | Similarity | 7k | 1.5k | 1.4k | 1 | Pearson/Spearman corr |

