# OpenReview forum: "HomoDistil: Homotopic Task-Agnostic Distillation of Pre-trained Transformers"
_ICLR.cc/2023/Conference — ICLR 2023 poster_

### Official Review · Reviewer_1n6p · 2022-10-24

**Confidence:** 3
**Correctness:** 4
**Technical Novelty And Significance:** 2
**Empirical Novelty And Significance:** 3
**Recommendation:** 8

**Clarity, Quality, Novelty And Reproducibility:**

The paper is clear and well-written, with minor grammatical issues. The experiments are run five times, and medians are reported: I would have appreciated also seeing means and standard deviations. The method itself combines existing technique in a new setting, thus there is novelty, albeit somewhat limited.

Will the code be made available should the paper be accepted? This would obviously help with reproducibility.

**Strength And Weaknesses:**

Strengths:
* An efficient distillation method is proposed that leverages both KD and pruning
* Experiments are realistic, and show the superiority of the proposed method compared to the SOTA baselines
* Latent space mapping/alignment is done elegantly via a learned weight matrix

Weaknesses:
* Reported medians over 5 runs, no standard deviations: how robust is the method?
* Table 2 could be presented visually

**Summary Of The Paper:**

The authors propose a task-agnostic knowledge distillation method for NLP models based on iterative pruning combined with distillation. A novel distillation loss is proposed, that performs differentiable matching of the latent representations. The proposed distillation method is compared to a selection of baselines, and is shown to perform well.

**Summary Of The Review:**

Overall, I think this is a useful, practical paper, showing how existing methods and ideas can be combined to achieve higher quality solutions. Compressing huge models is an important task, since many resource-constrained settings exist that currently can’t leverage the modern NLP models. The authors should clarify if the latent representation alignment via a differentiable weight matrix is their idea: this is obviously an excellent way of discovering the alignment, I have not encountered it in the literature before. However, if the idea is borrowed, the sources must be stated clearly.

See below for additional suggested corrections:

Page 4: “the loss for continual pre-training the student model on the open-domain data” -> the loss for continual pre-training of the student model on the open-domain data
Page 4: “consists both the distillation…” -> consists of both…

Medians of 5 runs are reported - what about the corresponding standard deviations? Perhaps Table 2 can be represented visually (e.g. box-and-whisker plots)?

Regarding the matching rule: you propose a differentiable weighted matching scheme, i.e. the matching is determined via training. Is my understanding correct? Has similar matching been applied in the NLP domain before? Please indicate the sources.

---

> ### Author Response · Authors · 2022-11-18
> **Response to Reviewer 1n6p (Part 1)**
>
> ### Reported medians over 5 runs, no standard deviations: how robust is the method? Will the code be made available should the paper be accepted?
>
> The standard deviations for Table 2 in the paper (**Please see Appendix A.7 for more details**):
>
> |                 | MNLI-m Std. | MNLI-mm Std. | QQP-Acc Std. | QQP-F1 Std. | QNLI Std. | SST-2 Std. | CoLA Std. | RTE Std. | MRPC-Acc Std. | MRPC-F1 Std. | STS-B P Std. | STS-B S Std. |
> |-----------------|-------------|--------------|--------------|-------------|-----------|------------|-----------|----------|---------------|--------------|--------------|--------------|
> | HomoBERT-base   |        0.13 |         0.20 |         0.09 |        0.12 |      0.34 |       0.24 |      1.72 |     0.93 |          1.17 |         0.83 |         0.15 |         0.18 |
> | HomoBERT-small  |        0.23 |         0.14 |         0.08 |        0.20 |      0.14 |       0.27 |      2.49 |     1.29 |          0.25 |         1.43 |         0.23 |         0.25 |
> | HomoBERT-xsmall |        0.14 |         0.12 |         0.08 |        0.10 |      0.24 |       0.61 |      1.32 |     1.29 |          0.95 |         0.66 |         0.27 |         0.28 |
> | HomoBERT-tiny   |        0.16 |         0.29 |         0.11 |        0.13 |      0.29 |       0.16 |      1.26 |     1.26 |          1.05 |         0.62 |         0.19 |         0.22 |
>
> The standard deviations for Table 3 in the paper:
>
> |                 | SQuaD 1.1EM Std.  | SQuaD 1.1 F1 Std. | SQuaD 2.0 EM Std. | SQuaD 2.0 F1 Std. |
> |-----------------|-------------------|-------------------|-------------------|-------------------|
> | HomoBERT-small  |              0.24 |              0.19 |              0.37 |              0.37 |
> | HomoBERT-xsmall |              0.15 |              0.19 |              0.62 |              0.59 |
> | HomoBERT-tiny   |              0.29 |              0.31 |              0.78 |              0.72 |
>
> As promised in the abstract, we will make our implementation and checkpoints publicly available for our community.
>
> ### The authors should clarify if the latent representation alignment via a differentiable weight matrix is their idea.
>
> This idea was initially proposed by FitNets (Romero, Adriana, et al. 2014). They chose a "hint layer" from the teacher and the student models, respectively, and aligned their latent representations via a differentiable weight matrix. In Transformer-based language models, TinyBERT (Jiao et al, 2019) has adopted a similar idea. They proposed to align the latent representations of all layers via one differentiable weight matrix. Our approach further builds upon TinyBERT's approach in two ways: 1) We align the latent representations at each layer with a layer-specific differentiable weight matrix. 2) This layer-specific matrix is initialized as a square matrix with the teacher's dimension, then its output dimension is gradually pruned to match the student's changing dimension throughout the distillation process.
>
> *References:*
>
> *Romero, Adriana, et al. "Fitnets: Hints for thin deep nets." arXiv preprint arXiv:1412.6550 (2014).*
>
> *Jiao, Xiaoqi, et al. "Tinybert: Distilling bert for natural language understanding." arXiv preprint arXiv:1909.10351 (2019).*
>
> ### Typos and visualization suggestions.
>
> Thanks for the nice suggestions! We have revised the paper accordingly.

---

> > ### Comment · Reviewer_1n6p · 2022-11-23
> > **Thank you for the clarification**
> >
> > Thank you, I appreciate the clarifications and the references! I assume these have been mentioned in the paper, too.

---

> > > ### Author Response · Authors · 2022-11-27
> > > **Thanks for your comments**
> > >
> > > Thank you for taking the time to read our responses! Please consider raising your score if these responses have addressed your concerns, or let us know if you have any further concerns so we can discuss them!

---

> ### Author Response · Authors · 2022-11-18
> **Response to Reviewer 1n6p (Part 2)**
>
> ### The method itself combines existing techniques in a new setting, thus there is novelty, albeit somewhat limited.
>
> Our proposed method is not a simple, straightforward combination of pruning and distillation. In contrast, it has a better performance than distillation alone (Table 2, 3 in our paper), pruning alone (Table 4 in our paper), or their straightforward combination (Figure 2 in our paper). Below we list our contributions from two main perspectives:
>
> **Research Significance.** We are the first to show that pruning can be leveraged to alleviate the teacher-student capacity gap issue in distillation:
>
> 1. Our observations provide a deeper understanding and some new insights for distillation. For example, after combined with pruning, distillation under large compression ratios can achieve significantly better performance than before (Table 2, 3 in our paper); by pruning with a longer schedule, we can maintain a smaller prediction discrepancy and achieve better performance (Figure 2 in our paper).
> 2. Our results are of great practical significance and implications. As powerful models are growing rapidly in scale, our results indicate the possibility of leveraging a much larger and more powerful teacher to produce a very small student with minimal loss of accuracy and point out a new research direction for achieving this goal.
>
> **Technical Novelties.** Similar to many works that build upon existing techniques, our technical novelties lie in “where” and “how” we combine existing techniques. Below we compare our method with the relevant methods we’ve discussed in Section 6 of the paper: DynaBERT (Hou et al., 2020), which first prune then distill; SparseBERT (Xu et al., 2021) and CoFi (Xia et al., 2022), both of which simultaneously prune and distill:
>
> 1. We consider the task-agnostic setting while existing works focus on the task-specific setting:
>     - In the task-specific setting, existing works need to prune a model for each individual task. In contrast, a single task-agnostic model can be fine-tuned for any task at a low cost.
>     - In the task-specific setting, pruning incurs an inevitable loss of task-relevant pre-training knowledge that may not be present in the fine-tuning data. Therefore, existing works leave the word embeddings untouched (e.g., take up around $20$ million parameters in BERT-base). In contrast, this problem does not exist in the task-agnostic setting. This allows us to prune the word embeddings and produce a smaller model that is more suitable for edge devices (e.g., around $15$ million parameters).
>
> 2. Since it more difficult to maintain a small discrepancy over massive amount of pre-training data in task-agnostic distillation, we propose the following strategies:
>     - We initialize the student with the teacher to maintain a small discrepancy in the early stage. In contrast, DynaBERT initializes the student with a target-size model. SparseBERT and CoFi initialize the student with pre-trained weights while the teacher with fine-tuned weights.
>     - We simultaneously prune and distill and allow interactions between them. We prune based on a newly proposed "column sensitivity" to make the pruning operation ''distillation-aware''. Specifically, we select rows and columns to prune based on their contributions to the distillation loss. In contrast, DynaBERT treats pruning and distillation as two independent operations by first pruning then distilling the subnetwork; SparseBERT prunes based on the weight magnitude without considering the influence on the distillation loss.
>
> 3. We propose strategies to ease the model deployment in practice while maintaining the model accuracy:
>     - We prune at the granularity of rows and columns. This granularity is sufficiently small to control the increment in discrepancy while maintaining the practical benefits of structured pruning. In contrast, DynaBERT and CoFi prune at a large granularity, which compromises the accuracy; SparseBERT conducts unstructured pruning, which compromises the practical benefits.
>     - We control the layer width by enforcing a local sparsity constraint. In contrast, SparseBERT and CoFi have no control over the layer width, which might result in wide matrices as memory bottlenecks.

---

### Official Review · Reviewer_Y2q6 · 2022-10-25

**Confidence:** 2
**Correctness:** 3
**Technical Novelty And Significance:** 2
**Empirical Novelty And Significance:** 2
**Recommendation:** 6

**Clarity, Quality, Novelty And Reproducibility:**

The paper is clear and easy to follow, the novelty is limited. No code is provided, but the experiment seems to be reproducible.

**Strength And Weaknesses:**

The proposed method is simple yet effective, good performance are obtained according to experiments;

The paper is easy to follow;

However, the proposed method has limited novelty, it is based on some existing methods;

It would be nice if the authors can provide a comprehensive comparison of computation cost/time between the proposed method and previous methods;

Why results of model with 66M parameters is not reported in Table 3?

Please adjust the position of Table 2, Table 3 and the paragraph between the two tables.

Some related works are suggested to be compared.

[1]. Multi-Granularity Structural Knowledge Distillation for Language Model Compression. Chang Liu, Chongyang Tao, Jiazhan Feng, Dongyan Zhao

[2]. BERT Learns to Teach: Knowledge Distillation with Meta Learning. Wangchunshu Zhou, Canwen Xu, Julian McAuley.

[3]. LRC-BERT: Latent-representation Contrastive Knowledge Distillation for Natural Language Understanding. Hao Fu, Shaojun Zhou, Qihong Yang, Junjie Tang, Guiquan Liu, Kaikui Liu, Xiaolong Li.

[4]. MixKD: Towards Efficient Distillation of Large-scale Language Models. Kevin J Liang, Weituo Hao, Dinghan Shen, Yufan Zhou, Weizhu Chen, Changyou Chen, Lawrence Carin.


**Summary Of The Paper:**

This paper propose HomoDistil, a method which combines knowledge distillation and pruning to obtain efficient networks.

**Summary Of The Review:**

Although the proposed method is simple and seems effective, I lean towards rejection because of the limited novelty.

---

> ### Author Response · Authors · 2022-11-18
> **Response to Reviewer Y2q6 (Part 1)**
>
> ### Some related works are suggested to be compared.
>
> Tables 2 and 3 in our paper have compared HomoDistil with the commonly used task-agnostic distillation methods. We further compare HomoDistil with the commonly used task-specific distillation methods (**Please see Appendix A.4 for more details**):
>
> | Method                              | Setting       | Params (million) | MNLI-m | MNLI-mm | QQP  | QNLI | SST-2 | CoLA | RTE  | MRPC | STS-B | Avg  |
> |-------------------------------------|---------------|------------------|--------|---------|------|------|-------|------|------|------|-------|------|
> | PKD_6 (Sun et al., 2019)            | Task-Specific | 66               | 81.3   | -       | 88.4 | 88.4 | 91.3  | 45.5 | 66.5 | 85.7 | 86.2  | 79.2 |
> | BERT-of-Theseus_6 (Xu et al., 2020) | Task-Specific | 66               | 82.3   | -       | 89.6 | 89.5 | 91.5  | 51.1 | 68.2 | 89.0 | 88.7  | 81.2 |
> | MixKD_6 (Liang et al., 2020)        | Task-Specific | 66               | 82.5   | -       | 90.8 | 88.8 | 92.1  | -    | 67.9 | 84.1 | -     | -    |
> | DynaBERT_6 (Hou et al., 2020)       | Task-Specific | 66               | 83.7   | 84.6    | 91.1 | 90.6 | 92.7  | 54.6 | 66.1 | 85.0 | 88.6  | 81.6 |
> | ProKT_6 (Shi et al., 2021)          | Task-Specific | 66               | 82.8   | 83.2    | 90.9 | 89.7 | 91.3  | 54.3 | 68.4 | 86.3 | 88.9  | 81.6 |
> | MetaDistil_6 (Zhou et al., 2022)    | Task-Specific | 66               | 83.5   | 83.8    | 91.0 | 90.4 | 92.3  | 58.6 | 69.4 | 86.8 | 89.1  | 82.6 |
> | HomoBERT-base                       | Task-Agnostic | 65               | 84.2   | 84.3    | 91.2 | 90.7 | 92.7  | 55.9 | 77.6 | 89.0 | 89.2  | 83.8 |
>
> LRC-BERT shows 14.5M results only, so we list the comparison here. Note LRC-BERT is a two-stage method and may be unfair to directly compare HomoDistil with.
>
>
> | Method                     | Setting                       | Params (million) | MNLI-m | MNLI-mm | QQP  | QNLI | SST-2 | CoLA | RTE  | MRPC | STS-B | Avg  |
> |----------------------------|-------------------------------|------------------|--------|---------|------|------|-------|------|------|------|-------|------|
> | HomoBERT-xsmall            | Task-Agnostic                 | 15.6             |   81.5 |    81.8 | 90.0 | 88.0 |  90.3 | 40.8 | 71.5 | 87.7 |  88.0 | 79.7 |
> | LRC-BERT (Fu et al., 2021) | Task-Agnostic + Task-Specific | 14.5             |   83.4 |    83.5 | -    | -    | -     | 50.0 | -    | 89.0 | -     | -    |
>
> *References:*
>
> *Sun, Siqi, et al. "Patient knowledge distillation for bert model compression." arXiv preprint arXiv:1908.09355 (2019).*
>
> *Xu, Canwen, et al. "Bert-of-theseus: Compressing bert by progressive module replacing." arXiv preprint arXiv:2002.02925 (2020).*
>
> *Liang, Kevin J., et al. "MixKD: Towards efficient distillation of large-scale language models." arXiv preprint arXiv:2011.00593 (2020).*
>
> *Hou, Lu, et al. "Dynabert: Dynamic bert with adaptive width and depth." Advances in Neural Information Processing Systems 33 (2020): 9782-9793.*
>
> *Shi, Wenxian, et al. "Follow your path: a progressive method for knowledge distillation." Joint European Conference on Machine Learning and Knowledge Discovery in Databases. Springer, Cham, 2021.*
>
> *Zhou, Wangchunshu, Canwen Xu, and Julian McAuley. "BERT learns to teach: Knowledge distillation with meta learning." Proceedings of the 60th Annual Meeting of the Association for Computational Linguistics (Volume 1: Long Papers). 2022.*
>
> *Fu, Hao, et al. "LRC-BERT: latent-representation contrastive knowledge distillation for natural language understanding." Proceedings of the AAAI Conference on Artificial Intelligence. Vol. 35. No. 14. 2021.*
>
>
>
> ### Please adjust the position of Table 2, Table 3 and the paragraph between the two tables.
>
> Thanks for pointing that out! We have revised the paper accordingly.

---

> ### Author Response · Authors · 2022-11-18
> **Response to Reviewer Y2q6 (Part 2)**
>
> ### The proposed method has limited novelty, it is based on some existing methods.
>
> Our proposed method is not a simple, straightforward combination of pruning and distillation. In contrast, it has a better performance than distillation alone (Table 2, 3 in our paper), pruning alone (Table 4 in our paper) or their straightforward combination (Figure 2 in our paper). Below we list our contributions from two main perspectives:
>
> **Research Significance.** We are the first to show that pruning can be leveraged to alleviate the teacher-student capacity gap issue in distillation:
>
> 1. Our observations provide a deeper understanding and some new insights for distillation. For examples, after combined with pruning, distillation under large compression ratios can achieve significantly better performance than before (Table 2, 3 in our paper); by pruning with a longer schedule, we can maintain a smaller prediction discrepancy and achieve better performance (Figure 2 in our paper).
> 2. Our results are of great practical significance and implications. As powerful models are growing rapidly in scales, our results indicate the possibility of leveraging a much larger and more powerful teacher to produce a very small student with minimal loss of accuracy, and point out a new research direction for achieving this goal.
>
> **Technical Novelties.** Similar to many works that build upon existing techniques, our technical novelties lie in “where” and “how” we combine existing techniques. Below we compare our method with the relevant methods we’ve discussed in Section 6 of the paper: DynaBERT (Hou et al., 2020), which first prune then distill; SparseBERT (Xu et al., 2021) and CoFi (Xia et al., 2022), both of which simultaneously prune and distill:
>
> 1. We consider the task-agnostic setting while existing works focus on the task-specific setting:
>     - In the task-specific setting, existing works need to prune a model for each individual task. In contrast, a single task-agnostic model can be fine-tuned for any task at a low cost.
>     - In the task-specific setting, pruning incurs an inevitable loss of task-relevant pre-training knowledge that may not be present in the fine-tuning data. Therefore, existing works leave the word embeddings untouched (e.g., take up around $20$ million parameters in BERT-base). In contrast, this problem does not exist in the task-agnostic setting. This allows us to prune the word embeddings and produce a smaller model that is more suitable for edge devices (e.g., around $15$ million parameters).
>
> 2. Since it more difficult to maintain a small discrepancy over massive amount of pre-training data in task-agnostic distillation, we propose the following strategies:
>     - We initialize the student with the teacher to maintain a small discrepancy in the early stage. In contrast, DynaBERT initializes the student with a target-size model. SparseBERT and CoFi initialize the student with pre-trained weights while the teacher with fine-tuned weights.
>     - We simultaneously prune and distill and allow interactions between them. We prune based on a newly proposed "column sensitivity" to make the pruning operation ''distillation-aware''. Specifically, we select rows and columns to prune based on their contributions to the distillation loss. In contrast, DynaBERT treats pruning and distillation as two independent operations by first pruning then distilling the subnetwork; SparseBERT prunes based on the weight magnitude without considering the influence on the distillation loss.
>
> 3. We propose strategies to ease the model deployment in practice while maintaining the model accuracy:
>     - We prune at the granularity of rows and columns. This granularity is sufficiently small to control the increment in discrepancy while maintaining the practical benefits of structured pruning. In contrast, DynaBERT and CoFi prune at a large granularity, which compromises the accuracy; SparseBERT conducts unstructured pruning, which compromises the practical benefits.
>     - We control the layer width by enforcing a local sparsity constraint. In contrast, SparseBERT and CoFi have no control over the layer width, which might result in wide matrices as memory bottlenecks.

---

> ### Author Response · Authors · 2022-11-18
> **Response to Reviewer Y2q6 (Part 3)**
>
> ### Why results of model with 66M parameters is not reported in Table 3?
>
> We did not report the 66M results in Table 3 because we were unable to reproduce the number of MiniLM from the released checkpoints. Despite this, we present the 66M results in the Table below. "(Paper)" results are from the original paper. All other results are fine-tuned from the released checkpoints and are the median of five random seeds.
>
> We can observe that the gain is noticeable but not as prominent as in the ~15M case, which is consistent with the observation in Table 2 of the paper. We conjecture it is because combining pruning with distillation brings more benefits under a larger capacity gap (e.g., the ~15M case), while the compression ratio of 50% is moderate and can be handled well by distillation alone.
>
> |                                  | SQuAD 1.1EM | Std. | SQuAD 1.1 F1 | Std. | SQuAD 2.0 EM | Std. | SQuAD 2.0 F1 | Std. | Avg  |
> |----------------------------------|-------------|------|--------------|------|--------------|------|--------------|------|------|
> | BERT-base (Teacher)              |        81.7 | 0.33 |         88.9 | 0.18 |         73.4 | 0.65 |         76.7 | 0.64 | 82.8 |
> | DistilBERT_6                     |        78.2 | 0.12 |         86.0 | 0.07 |         66.5 | 0.37 |         69.5 | 0.42 | 77.7 |
> | TinyBERT-GD_6                    |        73.9 | 0.74 |         83.2 | 0.65 |         69.2 | 0.50 |         73.0 | 0.43 | 78.1 |
> | MiniLMv1_6 (Paper)               | -           | -    | -            | -    | -            | -    |         76.4 | -    | -    |
> | MiniLMv1_6 (Paper)               | -           | -    | -            | -    | -            | -    |         76.3 | -    | -    |
> | MiniLMv2_6                       |        80.2 | 0.24 |         88.2 | 0.16 |         72.5 | 0.17 |         75.6 | 0.17 | 81.9 |
> | HomoBERT-base                    |        80.5 | 0.22 |         88.4 | 0.17 |         72.4 | 0.25 |         75.8 | 0.24 | 82.1 |
>
>
> ### It would be nice if the authors can provide a comprehensive comparison of computation cost/time between the proposed method and previous methods.
>
> We compare the computational costs of HomoDistil and the baseline methods during inference in the table below (**Please see Appendix A.6 for more details**). We profile the inference time and the number of FLOPs (embedding excluded) during the forward pass using [pytorch profiler](https://pytorch.org/tutorials/recipes/recipes/profiler_recipe.html). We conduct the measurements on the GLUE development set with a batch size of $128$ and a maximum sequence length of $128$ on one Nvidia A100 GPU. We compute the averaged time and FLOPs over all batches. The speedup is computed with respect to BERT-base.
>
> As can be observed, HomoDistil achieves some inference speedup and FLOPs reduction, but not as much as the other models under a similar parameter budget. This is because HomoDistil allocates a higher budget to the backbone parameters and a lower budget to the embedding parameters. We remark that HomoDistil focuses more on saving memory and storage costs. Given the same parameter budget, HomoDistil achieves better accuracy and enjoys the same (or more) storage benefits than the distilled (or structured pruned) models.
>
> |                     | Params (million) | Inference Speedup | Backbone #FLOPs |
> |---------------------|------------------|-------------------|-----------------|
> | BERT-base           |              109 |              1.00x |            1.00x |
> | DistilBERT_6        |               66 |              1.98x |            0.50x |
> | TinyBERT_6-GD       |               66 |              1.98x |            0.50x |
> | MiniLMv1_6          |               66 |              1.98x |            0.50x |
> | MiniLMv2_6          |               66 |              1.98x |            0.50x |
> | HomoBERT-base       |               65 |              1.30x |            0.56x |
> | BERT-small          |             28.6 |              4.77x |            0.15x |
> | CoFi-5%             |             28.4 |              5.16x |            0.05x |
> | TinyBERT_{3x384}-GD |             17.0 |              7.34x |            0.06x |
> | MiniLMv1_3          |             17.0 |              7.34x |            0.06x |
> | TinyBERT_{4x312}-GD |             14.5 |              6.28x |            0.07x |
> | HomoBERT-small      |             17.1 |              2.40x |            0.11x |
> | HomoBERT-xsmall     |             15.6 |              2.51x |            0.10x |
> | HomoBERT-tiny       |             14.1 |              2.55x |            0.09x |

---

> ### Comment · Reviewer_Y2q6 · 2022-11-26
> **Response**
>
> I have read the response from the authors, which is quite clear. I also checked the results from the Appendix.
> I will raise my score to 6, and hope the authors add more discussions about novelty and limitation in their future revision,

---

> > ### Author Response · Authors · 2022-11-27
> > **Thank you for your comments**
> >
> > Thank you for taking the time to read our responses! We will add more discussions about limitation and novelty in the next version.

---

### Official Review · Reviewer_na3M · 2022-10-25

**Confidence:** 2
**Correctness:** 3
**Technical Novelty And Significance:** 3
**Empirical Novelty And Significance:** 3
**Recommendation:** 6

**Clarity, Quality, Novelty And Reproducibility:**

The wrting is clear, the whole paper is easy to understand. I see the experiments are all conducted on public benchmarks.

**Strength And Weaknesses:**

strength:
* HomoDistill illustrates advantageous performance over other distillation methods like TinyBERT.
* The combination of pruning and distillation in task-agnostic setting seems novel.

weakness:
* A comparison with other prune & distill methods such as (Xu et al., 2021; Xia et al., 2022) would make this submission more complete.
* Please also discuss the difference between HomoDistill and [1], which presents a similar method for vision tasks, where a teacher model is slimmed into a small student model while distilling it.

[1] Slimmable Neueral Networks, ICLR 2019.

**Summary Of The Paper:**

This paper tackles with a problem that when distilling a small student model using a large pre-trained transformer, the “capacity gap” between the student and the teacher often hinders the teacher from transferring good performance to the student. The HomoDistill method proposed by this paper investigates to progressively prune the teacher model into a small student model, while in the same time distilling it using the original teacher model. Experiments show that the combination of pruning and distillation shows advantage on several NLU and QA tasks.

**Summary Of The Review:**

In general, I think HomoDistill is a novel method that combines pruning with distillation in a task-agnostic setting for NLP tasks. The experiments on NLU and QA benchmarks show the advantages of HomoDistill over other distillation methods. However, the paper lacks experimental comparison with other methods that combine pruning and distillation. Appending such comparison would make this paper more solid.

---

> ### Author Response · Authors · 2022-11-18
> **Response to Reviewer na3M (Part 1)**
>
> ### A comparison with other prune & distill methods such as (Xu et al., 2021; Xia et al., 2022) would make this submission more complete.
>
> The table below shows the performance comparison among HomoDistil, CoFi (Xia et al., 2022), and SparseBERT (Xu et al., 2021) on the GLUE benchmark. We compare the performance at a small model scale where these papers report their performances. HomoDistil achieves a noticeable gain over CoFi and a comparable performance with SparseBERT with nearly half of their sizes. **Please see Appendix A.5 for more details**.
>
> | Model                           | Params (million) | MNLI-m | MNLI-mm | QQP  | QNLI | SST-2 | CoLA | RTE  | MRPC | STS-B | Avg  |
> |---------------------------------|------------------|--------|---------|------|------|-------|------|------|------|-------|------|
> | CoFi-5%                         | 28.4             |   80.6 | -       | 90.1 | 86.1 |  90.6 | 35.6 | 64.7 | 82.6 |  83.1 | 76.7 |
> | SparseBERT-5%                   | 28.4             | -      | -       | -    | 90.6 | -     | 52.1 | 69.1 | 88.5 | -     | -    |
> | HomoBERT-xsmall                 | 15.6             |   81.5 |    81.8 | 90.0 | 88.0 |  90.3 | 40.8 | 71.5 | 87.7 |  88.0 | 79.7 |
>
> The two tables below present the method comparison among HomoDistil, CoFi, and SparseBERT from the distillation and pruning perspectives, respectively. **Please see Appendix A.5 for more details**.
>
> | Method     | Distillation Setting | Teacher Model      | Student Initialization      | Pruning Criterion        | Pruning Granularity      | Controllable Layer Width |
> |------------|----------------------|--------------------|-----------------------------|--------------------------|--------------------------|--------------------------|
> | CoFi       | Task-specific        | Fine-tuned weights | Pre-trained weights         | l0 regularization        | Layer, attention head, FFN, weight | No             |
> | SparseBERT | Task-specific        | Fine-tuned weights | Pre-trained weights         | Weight magnitude         | Weight                   | No                       |
> | HomoDistil | Task-agnostic        | Pre-trained weights| Pre-trained weights         | Column sensitivity       | Row, column              | Yes                      |
>
> We summarize the major differences below:
>
> **HomoDistil focuses on the task-agnostic setting.** In the task-specific setting, pruning incurs an inevitable loss of task-relevant pre-training knowledge that may not be present in the fine-tuning data. Therefore, existing works leave the word embeddings untouched (e.g., take up around $20$ million parameters in BERT-base). In contrast, this problem does not exist in the task-agnostic setting. This allows us to prune the word embeddings and produce a smaller model that is more suitable for edge devices (e.g., around $15$ million parameters). Furthermore, a task-specific model needs to be specifically pruned for each individual task, while a task-agnostic model can be fine-tuned for any task with a low cost.
>
> Since it is more difficult to maintain a small discrepancy over a massive amount of pre-training data in the task-agnostic setting, we propose the following strategies:
>
> - **HomoDistil initializes the student with the teacher.** To maintain a small discrepancy in the early stage, HomoDistil initializes the student with the teacher. In contrast, SparseBERT and CoFi initialize the student with pre-trained weights while the teacher with fine-tuned weights.
>
> - **HomoDistil simultaneously prunes and distills and allows interactions between them.** To maintain a small discrepancy throughout distillation, HomoDistil prunes based on a newly proposed "column sensitivity" to make the pruning operation ''distillation-aware''. Specifically, HomoDistil selects rows and columns to prune based on their contributions to the distillation loss. In contrast, SparseBERT prunes based on the weight magnitude without considering the influence on the distillation loss.
>
> - **HomoDistil prunes rows and columns.** The granularity of rows and columns is sufficiently small to control the increment in discrepancy while maintaining the practical benefits of structured pruning. In contrast, CoFi prunes at a large granularity, which compromises the accuracy; SparseBERT conducts unstructured pruning, which compromises the practical benefits.
>
> - **HomoDistil can control the layer width.** HomoDistil enforces a local sparsity constraint for each matrix, producing a model with consistent width in each layer. In contrast, SparseBERT and CoFi have no control over the layer width, which might result in wide matrices as memory bottlenecks.
>
> *References:*
>
> *Xia, Mengzhou, Zexuan Zhong, and Danqi Chen. "Structured pruning learns compact and accurate models." arXiv preprint arXiv:2204.00408 (2022).*
>
> *Xu, Dongkuan, et al. "Rethinking Network Pruning--under the Pre-train and Fine-tune Paradigm." arXiv preprint arXiv:2104.08682 (2021).*

---

> ### Author Response · Authors · 2022-11-18
> **Response to Reviewer na3M (Part 2)**
>
> ### Please also discuss the difference between HomoDistill and Slimmable Neueral Networks, which presents a similar method for vision tasks, where a teacher model is slimmed into a small student model while distilling it.
>
> We believe there are some misunderstandings. This paper is not directly relevant to distillation or pruning. Aside from sharing the motivation of model compression, Slimmable NN is very different from ours in both the problem setting and method: Given a set of target widths, e.g., [0.25, 0.5, 0.75], it trains a single full model by jointly training all target-width subnetworks in it, with each subnetwork maintaining an individual set of batch norm parameters. Then during inference, a subnetwork is selected from the shared model on the fly to execute based on the device's computational constraint.
>
> The method is not directly applicable to Transformers as batch norm statistics play an important role. However, its high-level idea, which is similar to ensemble learning, has been explored in Transformers. Existing works have shown that ensemble learning is effective if the output consistency among the subnetworks is regularized. Therefore, it is still unclear if the idea of this paper will be effective in Transformers.

---

> ### Author Response · Authors · 2022-11-27
> **Reviewer na3M: Did our responses address your concerns?**
>
> Please let us know if our responses addressed your concerns about the lack of comparison with relevant methods, and we are happy to answer any further questions!

---

> > ### Comment · Reviewer_na3M · 2022-12-01
> > **The responses address my concerns.**
> >
> > Thanks for the clarification. I see they are mentioned in the paper.

---

### Author Response · Authors · 2022-11-18
**We would like to thank all the reviewers!**

We would like to thank all the reviewers for providing thoughtful suggestions and for acknowledging HomoDistil is novel by combining pruning and distillation (na3M, 1n6p), shows superiority to the existing methods in multiple public benchmarks (na3M, Y2q6, 1n6p), and simple and efficient (Y2q6, 1n6p).

We hope our responses clear some of your concerns. Please don't hesitate if you have more questions and we are open to answer and discuss!

---

### Decision · Program_Chairs · 2023-01-20

**Decision:**

Accept: poster

**Justification For Why Not Higher Score:**

Methodological novelty is low since this paper is in some sense just combining iterative pruning with knowledge distillation.

**Justification For Why Not Lower Score:**

Thorough comparison against strong baselines indicate that this approach can meaningfully improve over existing work.

**Metareview: Summary, Strengths And Weaknesses:**

This paper studies task-agnostic knowledge distillation of pretrained language models. It combines iterative pruning with layer-wise knowledge distillation. The main strengths of the paper are its empirical contribution, improvements over strong baselines, and extensive analyses. The main weakness of the paper is low novelty from a methodological standpoint.

**Note From Pc:**

if the above contains the word "oral" or "spotlight" please see: "oral" presentation means -> notable-top-5% and "spotlight" means -> notable-top-25%. As stated in our emails, we are disassociating presentation type from AC recommendations